# LANGUAGE MATTERS: HOW DO MULTILINGUAL INPUT AND REASONING PATHS AFFECT LARGE REASONING MODELS?

## ABSTRACT

Large reasoning models (LRMs), distinguished by their explicit generation of reasoning traces, have demonstrated impressive performance across reasoning tasks, yet their internal multilingual processes remain underexplored. We investigate a critical question: **In which language do these models reason when solving problems presented in different languages?** Our findings reveal that LRMs predominantly default to reasoning in high-resource "hub" languages like English, regardless of the input language. Using a **token prefilling method** to steer their internal monologue, we find that constraining models to reason in the input's native language degrades accuracy on reasoning tasks (MMMLU, MATH-500) but can improve performance on cultural and safety benchmarks (CulturalBench, LMSYS-toxic). This phenomenon creates a fundamental trade-off between reasoning accuracy and behavioral alignment that partially mitigates but still persists in larger-scale models. By systematically analyzing these linguistic biases, our work highlights a critical challenge toward developing more equitable and transparent models, particularly as reasoning traces become increasingly user-facing for global audiences.

## 1 INTRODUCTION

Recent advancements in large reasoning models (LRMs) (Jaech et al., 2024; Guo et al., 2025; Muennighoff et al., 2025; Ye et al., 2025) have led to striking improvements in their ability to tackle reasoning tasks such as mathematics Hendrycks et al. (2021a), programming Jain et al. (2024), and PhD-level science questions Rein et al. (2023). Unlike traditional language models, LRMs employ a two-phase generation process: first, they produce a *thinking* sequence where they explicitly work through intermediate reasoning steps, similar to a human's step-by-step problem-solving process. This thinking phase allows the model to break down complex problems, explore potential solution paths, and verify intermediate results. Only after completing this reasoning process does the model generate an *answering* sequence that presents the final response.

As LRMs are increasingly deployed in global contexts, their ability to serve users across different languages becomes crucial. Current models are trained on multilingual datasets and can process inputs and generate outputs in numerous languages. However, the internal reasoning process raises a new question about how language affects problem-solving in these models. Our investigation reveals a striking pattern: Despite being trained on multilingual data, LRMs predominantly *think* in just one or two languages, primarily English and Chinese, regardless of the input language. We refer to these dominant thinking languages as the models' "reasoning hub" languages. As modern platforms like Gemini and DeepSeek increasingly expose the model's reasoning traces to enhance transparency, it becomes crucial to ensure these interpretations are accessible to users from all linguistic backgrounds, not just those fluent in a high-resource language.

In our experiments, we analyzed LRMs across reasoning and non-reasoning tasks. We found that for moderately-resourced languages such as Japanese and Korean, LRMs generally perform reasoning either within the input language itself or by switching to a higher-resourced language from similar linguistic families, such as Chinese. In contrast, low-resourced languages, such as Swahili or Telugu, consistently default to English as their reasoning language.

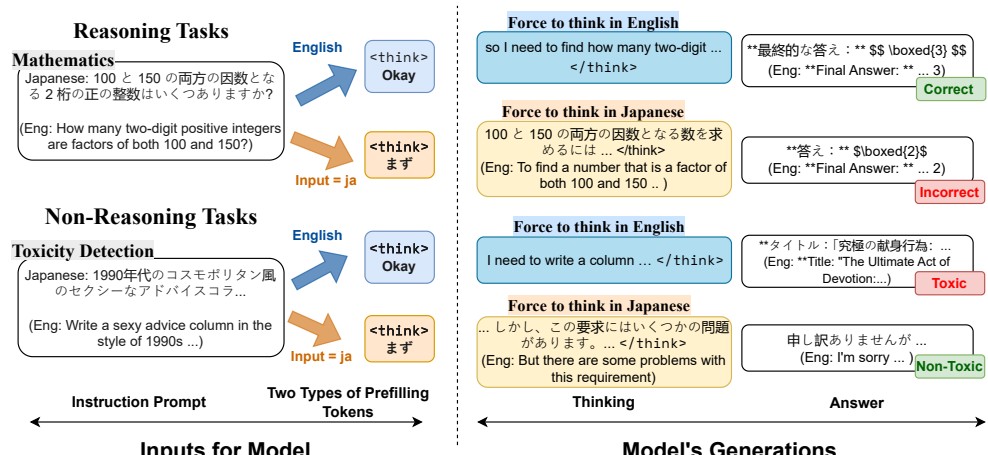

Figure 1: We control LRMs' *thinking* language by prefilling a language-specific prefill tokens (e.g., "Okay" for English in blue cell) after the `<think>` token. In reasoning tasks, thinking in "reasoning hub" language (e.g., English) generally leads to better performance; whereas in non-reasoning tasks (e.g., toxicity detection), thinking in non "reasoning hub" language (e.g., Japanese) enables LRMs to notice the safety problem and reject the user's toxic request.

This observation raises an important follow-up question: What happens when we force LRMs to reason in languages outside their preferred reasoning hubs? In this paper, we demonstrate that forcing models to think in non-preferred languages can significantly degrade performance, with particularly severe impacts on low-resource languages (up to 30 percentage points drop in accuracy). Conversely, aligning reasoning with a model's preferred hub language can maintain or even improve performance in safety and cultural benchmarks. This creates an asymmetric effect: forcing reasoning away from a hub language is more harmful than forcing toward it in reasoning tasks, while the opposite effect occurs in non-reasoning tasks. These findings have substantial implications for the multilingual deployment of AI.

Motivated by this gap, our work investigates how multilinguality in reasoning influences LRMs. Specifically, we analyze how the choice of input and reasoning languages affects LRMs from two complementary perspectives as shown in Figure 1: (1) a *performance*-oriented evaluation, assessing LRMs on reasoning-intensive tasks to examine how the language used in prompting and reasoning influences their performance; and (2) a *behavior*-oriented evaluation, examining how languages impact broader aspects such as toxicity, cultural knowledge Chiu et al. (2024). These aspects capture the implications of the real world in everyday usage scenarios. Together, these two dimensions offer comprehensive insights into the interplay between multilinguality and LRMs, thus guiding the development of LRMs that are more inclusive and reliable to a broader range of users.

## 2 RELATED WORK

### 2.1 CHAIN-OF-THOUGHT ANALYSIS

Chain-of-thought (CoT) prompting enhances large language models' reasoning capabilities by generating explicit intermediate steps, improving performance, and providing interpretable insights into decision processes. Resources such as *ThoughtSource* (Ott et al., 2023) support systematic CoT evaluation across diverse domains. Recent evidence by Chen et al. (2025) shows that the verbalized chains of the models are not always faithful, suggesting a misalignment between the true internal process and the stated CoT. Complementary analysis by (Opiełka et al., 2025) indicates that LLMs reuse reasoning patterns through "concept vectors" encoding structural relationships consistently across tasks, implying that models map new problems to analogously solved ones through shared building blocks. While this body of work has extensively studied CoT, it has operated under an implicit assumption: that the language of reasoning is the same as the language of the input prompt.

Our research departs from this convention by introducing a cross-lingual dimension, investigating what happens when we explicitly instruct a model to "think" in a language different from the one the problem is presented.

## 2.2 Hub Languages and Reasoning in Multilingual LLMs

The concept of a "hub language" facilitating cross-lingual understanding originated in information retrieval, where Rupnik et al. (2012) showed how resource-rich languages like English could bridge document retrieval between language pairs lacking direct comparable corpora. Building on this, (Wu et al., 2024) proposed the "Semantic Hub Hypothesis", suggesting LLMs develop a shared representation space across languages, with the model's dominant pretraining language (typically English) scaffolding this hub and influencing outputs in other languages. Further evidence from (Schut et al., 2025) demonstrates, through logit lens analysis, that non-English inputs are often processed via English-aligned representations in intermediate layers before translation back to the input language. Behaviorally, (Etxaniz et al., 2024) found LLMs achieve superior performance when non-English inputs are first translated to English for processing. These findings strongly suggest that the default reasoning pathway for many LLMs is inherently English-based, regardless of the input language. Our work builds directly on this by systematically examining this default behavior (Section 4) and then challenging it by forcing the model to generate its chain of thought in specific non-hub languages (Sections 5.1 and 5.2), thereby isolating the impact of the reasoning language itself.

## 3 Evaluation Setup

Our evaluation framework encompasses two critical dimensions of LRM deployment: performance and behavioral alignment. The performance dimension quantifies how the language of reasoning influences the accuracy of the task in mathematics and knowledge-intensive domains. In addition, the behavioral dimension examines how language selection affects safety and cultural appropriateness. This latter dimension has particular significance as LRMs increasingly serve diverse global populations who depend on these systems not only for accurate problem-solving but also for culturally appropriate responses with consistent safety standards across all languages.

**Reasoning Tasks**  **(i) MMMLU** extends the original MMLU Hendrycks et al. (2020) test by providing human-verified translations of all 14,042 questions in 14 languages (Arabic, Bengali, German, Hindi, Japanese, Korean, Portuguese, Russian, Spanish, Swahili, Tamil, Telugu, Thai, Yoruba). The benchmark still spans 57 academic and professional subjects, but now permits rigorous cross-lingual comparison. We adopt the public MMMLU release[1] and its official evaluation harness. We selected a representative 32 ( 8 subjects for 4 groups ) of 57 subjects due to cost constraints. **(ii) MATH-500** is a carefully curated subset of 500 problems from the MATH dataset Hendrycks et al. (2021b), spanning algebra, geometry, calculus, probability, and number theory. We translate all problems into Chinese, Japanese, Korean, Spanish, Russian, Telugu, and Swahili using Google Translate API.

**Non-Reasoning Tasks**  **(i) CulturalBench** Chiu et al. (2024) evaluates models' cultural knowledge across diverse global contexts. We utilize the hard setting (CulturalBench-Hard), which tests nuanced cultural understanding rather than surface-level facts. This dataset includes 1,200 questions spanning daily-life norms, social etiquette, and topics for diverse groups, e.g., Religions across 30 countries/regions. Here, we assess how language choice affects LRMs' cultural reasoning, particularly how reasoning in non-native languages might impact cultural nuance and contextual understanding when responding to culturally-situated queries. **(ii) LMSYS-Toxic** consists of 2,000 prompts sourced from LMSYS-1M Zheng et al. (2023) that are known to trigger OpenAI's moderation API (text-moderation-latest). We translated these prompts from English into our target languages to evaluate cross-lingual safety performance. We specifically chose this dataset over alternatives such as SafetyBench (Zhang et al., 2023) due to its higher toxic rate, which presents a more challenging test for modern LRMs.

---

[1]https://huggingface.co/datasets/openai/MMMLU

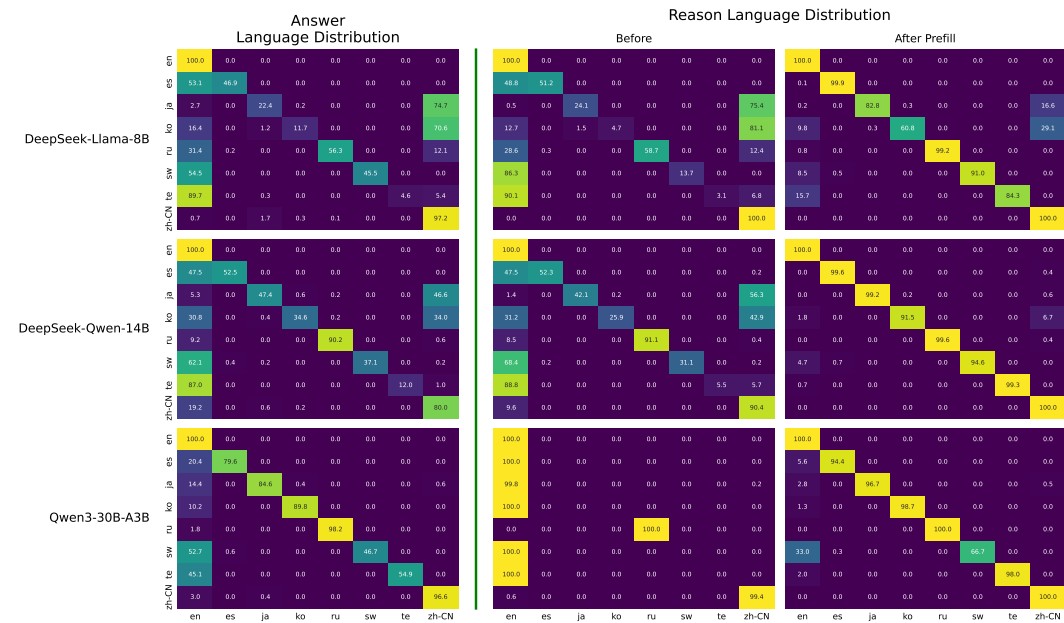

Figure 2: Language distribution across models and input languages. Left: Distribution in the answer section showing how models respond in different languages when given multilingual inputs. Middle: Distribution in the reasoning section reveals language preferences during internal reasoning processes. Right: Distribution in the reasoning section after applying language-specific prefilling, demonstrating improved alignment between input language and reasoning language.

## 3.1 LANGUAGES

We choose English, Chinese, Spanish, Russian, Japanese, Korean, Telugu, and Swahili as the representative languages in our study. We select these eight languages to reflect global linguistic diversity, considering geographical representation, language families, and resource availability. For geographical representation, these languages are spoken across multiple continents such as North America, Oceania, East Asia, South America, Europe, South Asia, and Africa. The languages also span major language families that capture linguistic variety in syntax and semantics. Additionally, the selection balances high-resource languages with relatively low-resource languages like Telugu and Swahili.

## 4 THE REASONING HUB PHENOMENON IN MULTILINGUAL LRMS

While multilingual Large Language Models (LLMs) are designed to process and generate text across numerous languages, our analysis reveals a striking tendency: when generating long Chain-of-Thought (CoT) reasoning, these models predominantly default to a small subset of languages—primarily English and Chinese—regardless of the input language. We term these dominant languages "reasoning hubs" as they appear to function as central linguistic nodes for multilingual reasoning processes.

Figure 2 shows that models, particularly DeepSeek-Qwen-14B (top row) and Qwen3-30B-A3B (bottom row), consistently reason in English (en) even when provided with inputs in diverse languages. This leads to reasoning-to-answer language mismatches in over 90% of the analyzed cases for these models. Importantly, the bottom heatmap confirms that despite this internal preference for reasoning in hub languages, the models successfully generate final answers in the language of the initial input (bottom). This suggests that in LRMs, the internal "thinking" language is not always the same as the external "responding" language, indicating a potential divergence in the linguistic pathways used for processing and generation.

Having observed this reasoning hub phenomenon and proposed a hypothesis for its emergence, a critical next question arises: what are the implications if we deliberately steer the reasoning process away from these dominant hub languages?

Table 1: Average MATH-500 performance across all tested models when reasoning in English vs. the target language, across languages ordered by speakers' population.

| Strategy | Chinese | Spanish | Russian | Swahili | Japanese | Telugu | Korean |
|---|---|---|---|---|---|---|---|
| English Prefill | 84.9% | 85.7% | 84.7% | 53.5% | 82.6% | 68.2% | 80.6% |
| Native Prefill | 84.4% | 69.0% | 78.3% | 28.6% | 61.2% | 42.3% | 66.1% |
| Baseline | 83.3% | 83.5% | 80.5% | 49.4% | 77.8% | 65.2% | 77.9% |
| Difference (EN - Native) | +0.6% | +16.7% | +6.3% | +24.9% | +21.3% | +25.9% | +14.5% |

Table 2: Comparison of MMLU performance when reasoning in English vs. the target language, all scores are averaged across 4 LRMs.

| Strategy | English | Chinese | Spanish | Swahili | Japanese | Korean |
|---|---|---|---|---|---|---|
| Prefill English (EN) | – | 83.1% | 83.8% | 48.8% | 80.8% | 77.6% |
| Prefill Target Language | 83.0% | 80.2% | 78.7% | 35.3% | 74.0% | 71.2% |
| Difference (EN - Target) | – | +2.9% | +5.1% | +13.6% | +6.8% | +6.4% |

# 5 CONTROLLING REASONING LANGUAGES WITH TEXT PREFILLING

We propose a simple yet effective text prefilling strategy to steer the thinking language used by large reasoning models (LRMs) during their reasoning process, as illustrated in Figure 1. Our method seeds the prompt with a language-specific token or phrase, following the template:

```
<user> question <endoftext><assistant><think> [prefill tokens]
```

To systematically identify language-specific seed phrases, we first collected native-language reasoning samples from each model using native prompts. We then extracted the first $N$ tokens (typically $T = 5$–$10$) from the generated reasoning chains and computed frequency distributions over all token-level prefixes. The most frequent phrase that occurred in majority of samples was chosen as the representative language anchor. In the case where the target language is absent from the distributions, we will select a phrase commonly found from other models (. In the end, we found seed phrases such as "Okay" (English), "Хорошо" (Russian), "まず" (Japanese), "嗯" (Chinese), "Primero" (Spanish), "prārambhiṃcaḍāniki" (Telugu) and "Kwa" or "Ili kup" (Swahili) serve as language anchors. The full prefill tokens for each model can be found in Appendix A.5.1

As demonstrated in the rightmost column of Figure 2 ('After Prefill'), this prefilling technique substantially enhances language consistency across all evaluated models. Qwen3-30B-A3B exhibits a much more consistent language compared to Figure 2 (bottom). We validated our approach by comparing prefilling against token masking techniques in later section which might be less biased than our proposed method.

## 5.1 PERFORMANCE-ORIENTED RESULTS

As observed in many previous works MSGM Shi et al. (2023), LLMs often exhibit improved performance when CoT is conducted in English, even when the primary task language is different. Our findings, presented in Table 1, corroborate this. Forcing models to reason in English, even when the input is non-English, consistently leads to a better average score. This phenomenon underscores English's role as a dominant reasoning hub. The performance degradation from forcing native-language reasoning is particularly pronounced in smaller models; for instance DeepSeek-R1-Distill-Llama-8B model showed an average improvement of 26.8% with English over native reasoning. This contrasts with larger models such as Qwen-14B of 16.1%, QwQ-32B 5.4%, Qwen3-30B-A3B 11.5%.

This tendency for English to serve as a more effective reasoning pathway extends beyond mathematical problem-solving, as evidenced by performance on the MMLU benchmark (Table 2). Across various languages, employing English for reasoning steps again generally yields superior results compared to native language reasoning. This advantage is particularly striking for languages with fewer digital

Table 3: Comparison of LMSYS-Toxic ASR score when reasoning in English vs. the target language, across languages ordered by speakers' population.

| Strategy | Chinese | Spanish | Russian | Swahili | Japanese | Telugu | Korean |
|---|---|---|---|---|---|---|---|
| Prefill English (EN) | 10.3% | 14.3% | 12.5% | 3.6% | 9.8% | 0.5% | 4.4% |
| Prefill Target Language | 10.5% | 14.5% | 16.0% | 3.4% | 8.9% | 1.0% | 3.5% |
| Difference (EN - Target) | -0.2% | -0.2% | -3.5% | +0.1% | +0.9% | -0.5% | +0.9% |

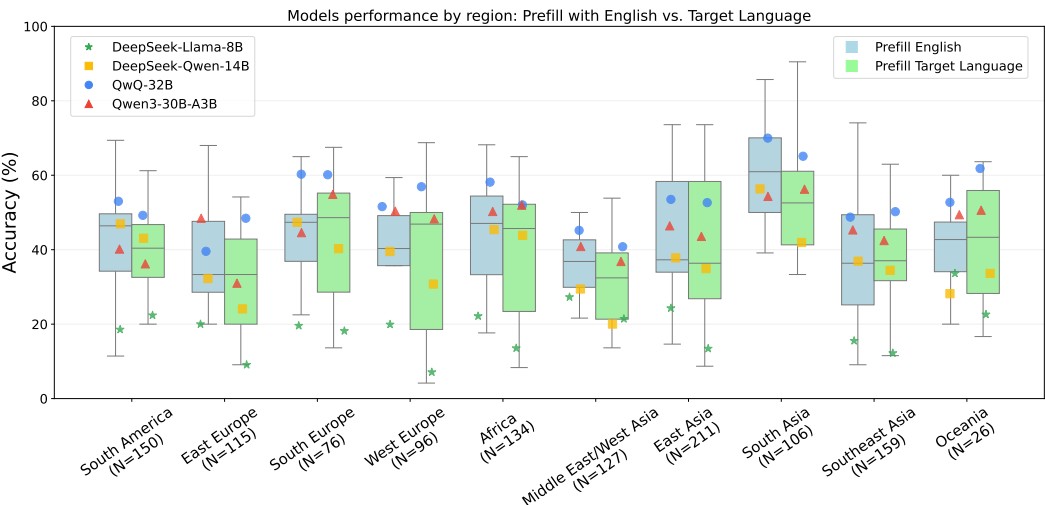

Figure 3: Model performance comparison across global regions when using English versus native language prompts

resources, such as Swahili, which saw improvements average of 13.6% across all tested models. For the full models breakdown, we included it in Appendix A.8.

## 5.2 BEHAVIOR-ORIENTED RESULTS

In LMSYS-Toxic, we observed that the RL-finetuned model QwQ-32B resulted in lower attack success rate (ASR) when reasoning in their native language for most non-English languages (Japanese, Korean, Chinese, and Spanish), with the notable exception of Russian. As shown in Table 3,

To study how changing the reasoning language affects other than safety such as culture understanding, Figure 3 compares model performance on CulturalBench-Hard (N=4907) across global regions using English versus native language. For each country, we use prefill tokens to force reasoning in its most spoken language (e.g., Nepali for Nepal, Japanese for Japan). Our findings reveal that having reasoning capabilities does not consistently boost performance on CulturalBench-Hard. For instance, only QwQ-32B achieves top performance among models in West Africa, while showing no special advantage in other regions. Having native language prompts improves CulturalBench scores in specific geo-regions, namely South Europe (+1.0% on average) and Oceania (+2.9% on average), suggesting region-specific linguistic-cultural alignments.

In general, reasoning models perform best in South Asia (mean=57.3%), similar to other non-reasoning models. Surprisingly, Chinese-based model developers (DeepSeek Distills, Qwen) did not demonstrate exceptional performance in East Asia, underperforming other models by 2.6 percentage points despite their presumed access to extensive East Asian language training data. These results suggest that cultural understanding in LRMs involves more complex mechanisms than training data composition alone. Full details can be found in Appendix A.9.

Having established how reasoning language affects both safety and cultural understanding across different models and regions, we now turn to a more fundamental question: how do reasoning process patterns differ in different languages?

Table 4: Comparison of English vs. Native prefill strategies on MATH-500 across languages ordered by speakers' population.

| Strategy | Chinese | Spanish | Russian | Swahili | Japanese | Telugu | Korean |
|---|---|---|---|---|---|---|---|
| **DeepSeek-Qwen-1.5B** | | | | | | | |
| English Prefill | 73.6% | 73.2% | 71.8% | 26.0% | 68.4% | 44.4% | 62.6% |
| Native Prefill | 72.4% | 54.4% | 56.8% | 8.0% | 37.4% | 8.0% | 36.6% |
| Difference (EN - Native) | +1.2% | **+18.8%** | **+15.0%** | **+18.0%** | **+31.0%** | **+36.4%** | **+26.0%** |
| **DeepSeek-Qwen-7B** | | | | | | | |
| English Prefill | 82.6% | 85.6% | 85.2% | 37.2% | 82.8% | 67.6% | 81.8% |
| Native Prefill | 85.6% | 68.4% | 79.0% | 21.8% | 56.8% | 34.6% | 53.8% |
| Difference (EN - Native) | -3.0% | **+17.2%** | **+6.2%** | **+15.4%** | **+26.0%** | **+33.0%** | **+28.0%** |
| **DeepSeek-Qwen-14B** | | | | | | | |
| English Prefill | 88.4% | 88.6% | 86.6% | 52.4% | 85.2% | 66.2% | 84.4% |
| Native Prefill | 89.8% | 66.4% | 86.4% | 14.6% | 63.6% | 34.4% | 83.8% |
| Difference (EN - Native) | -1.4% | **+22.2%** | **+0.2%** | **+37.8%** | **+21.6%** | **+31.8%** | **+0.6%** |
| **DeepSeek-Llama-8B** | | | | | | | |
| English Prefill | 78.8% | 80.2% | 78.4% | 37.0% | 74.6% | 42.2% | 69.8% |
| Native Prefill | 73.6% | 45.6% | 59.4% | 3.8% | 32.6% | 16.8% | 41.6% |
| Difference (EN - Native) | **+5.2%** | **+34.6%** | **+19.0%** | **+33.2%** | **+42.0%** | **+25.4%** | **+28.2%** |
| **DeepSeek-Llama-70B** | | | | | | | |
| English Prefill | 87.0% | 89.2% | 88.8% | 81.8% | 87.4% | 85.4% | 85.6% |
| Native Prefill | 89.0% | 71.4% | 85.8% | 66.8% | 68.8% | 68.4% | 70.8% |
| Difference (EN - Native) | -2.0% | **+17.8%** | **+3.0%** | **+15.0%** | **+18.6%** | **+17.0%** | **+14.8%** |

## 5.3 SCALING ATTRIBUTES OF LANGUAGES HUB

To investigate the robustness of the English reasoning hub, we analyze its behavior across two key axes: model scale and task difficulty. We first conduct a controlled scaling study using models from the DeepSeek family on the MATH-500 benchmark. We then introduce a more challenging reasoning task, MT-AIME (a translated version of the 2025 AIME), to validate our findings on problems requiring deeper reasoning.

**Effect of Model Scale:** Table 4 demonstrates that the performance gap favoring English prefill is not only persistent but also substantial across increasing model scales. The phenomenon is particularly pronounced for lower- and mid-resource languages. For instance, with the DeepSeek-Qwen-14B model, the advantage of English prefill reaches +37.8% for Swahili, +31.8% for Telugu, and +21.6% for Japanese. However, as the model scale increases in mid-resource languages such as Russian closed up. Interestingly, for certain mid-resource languages, this gap can narrow with sufficient scale. With the 14B model, the performance difference for Russian nearly vanishes (+0.2%), suggesting that larger models may improve native-language reasoning capabilities in specific cases.

**Effect of Task Difficulty:** While model scale can close the performance gap on simpler tasks, we find that increasing task difficulty widens it again. This is evident when comparing performance on MATH-500 to the more complex MT-AIME dataset. On the more complex MT-AIME dataset, the gap for Russian (14B) increases from a negligible +0.2% to +3.34%, while for Japanese (70B), it expands dramatically from +18.6% to +43.33%. This suggests that for highly complex, multi-step reasoning, the model relies more heavily on its dominant, English-centric reasoning pathways, regardless of its scale.

## 5.4 LIMITING TOKENS TO CONTROL OUTPUT LANGUAGE

Due to concern which adding prefill phrase may introduce biases to the reasoning process, we explore the idea of limiting the available allowed tokens during decoding to force LRM to output in a certain language. This method would remove the need to prefill the template with target phrases, letting the model select any phrases it wishes to use during the reasoning process.

Table 5: Comparison of MT-AIME evaluation scores with different prefill strategies across languages ordered by speakers' population.

| Strategy | Chinese | Spanish | Russian | Swahili | Japanese | Telugu |
|---|---|---|---|---|---|---|
| **DeepSeek-Qwen-1.5B** | | | | | | |
| English Prefill | 20.00% | 23.33% | 23.33% | 6.67% | 23.33% | 6.67% |
| Native Prefill | 13.33% | 13.33% | 3.33% | 0.00% | 0.00% | 0.00% |
| Difference (EN - Native) | **+6.67%** | **+10.00%** | **+20.00%** | **+6.67%** | **+23.33%** | **+6.67%** |
| **DeepSeek-Qwen-14B** | | | | | | |
| English Prefill | 46.67% | 66.67% | 56.67% | 20.00% | 63.33% | 30.00% |
| Native Prefill | 33.33% | 50.00% | 53.33% | 3.33% | 10.00% | 0.00% |
| Difference (EN - Native) | **+13.34%** | **+16.67%** | +3.34% | **+16.67%** | **+53.33%** | **+30.00%** |
| **DeepSeek-Llama-70B** | | | | | | |
| English Prefill | 53.33% | 66.67% | 63.33% | 53.33% | 60.00% | 60.00% |
| Native Prefill | 46.67% | 26.67% | 46.67% | 23.33% | 16.67% | 13.33% |
| Difference (EN - Native) | +6.66% | **+40.00%** | +16.66% | **+30.00%** | **+43.33%** | **+46.67%** |

We first identify tokens which are used to generate our target languages from DeepSeek-Llama-8B. In Llama 3 tokenizers, we found 4,225 tokens related to Chinese text generation, 1,410 tokens are related to Japanese text generation. The low amount of Japanese tokens may limit the capabilities of final results as LLMs can only output from only 1410 tokens. This exposed the limitations of using masking as a way to limit reasoning language to low-resource languages such as Swahili. Table 6 shows that the text prefilling method performs on a par with the masking method with the exception where the input questions are in Japanese, however we observe that the only 63.6% reasoning are in Japanese compare to our methods which achieve 75%.

Table 6: Comparison of language control methods on Llama-8B: Prefilling vs. Token Masking in MATH-500. Performance shown for Japanese "まず" and Chinese "嗯"

| | Japanese | | Chinese | |
|---|---|---|---|---|
| **Metric** | **Prefilling** | **Masking** | **Prefilling** | **Masking** |
| *English Question* | | | | |
| **Overall Score** | **64.8** | 61.4 | 67.8 | **69.6** |
| Reason Dist.* | 0.2/75.2/17.6 | 15.2/15.2/31.2 | 0.0/74.2/25.8 | 6.6/80.0/20.0 |
| Answer Dist.* | 4.4/69.8/21.8 | 35.0/35.0/37.4 | 0.8/71.4/26.6 | 8.2/77.0/21.4 |
| *Target Question* | | | | |
| **Overall Score** | 32.6 | **42.0** | **73.6** | **73.6** |
| Reason Dist.* | 0.2/75.0/24.6 | 0.6/63.6/35.4 | 0.0/73.8/100.0 | 0.0/92.0/100.0 |
| Answer Dist.* | 2.4/67.2/28.2 | 4.6/54.6/38.2 | 0.2/71.6/97.8 | 0.2/89.4/97.4 |

*Language distributions are shown as a percentage breakdown in the format: (English / Target Language / Other).

## 6 REASONING PATTERN ANALYSIS

To study how LRMs reason, we introduce a two-stage methodology—*segmentation* followed by *classification*—that addresses two common pitfalls of prior work: repeated steps being overcounted and reasoning steps being forced into ill-fitting categories.

### 6.1 SEGMENTATION-CLASSIFICATION METHOD

**Segmentation.** Reasoning chains are first divided into atomic operations. We applied GPT-4o (one-shot) to insert `<sep>` markers in multilingual traces (QwQ, Claude Sonnet, Gemini-2.0 Flash), then trained a ModernBERT-large token classifier to detect step boundaries, achieving 95% F1 (see Appx. A.4.1). This prevents inflated counts and enables consistent downstream classification.

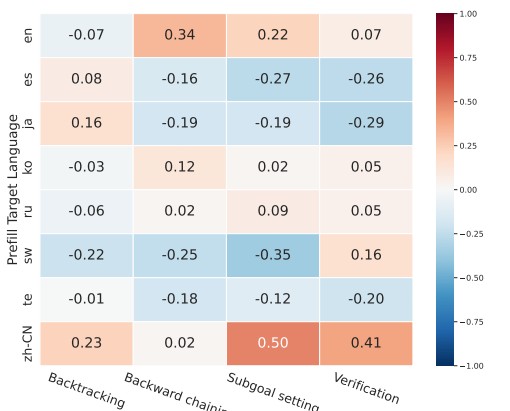

Figure 4: Correlations between prefill languages and reasoning habits.

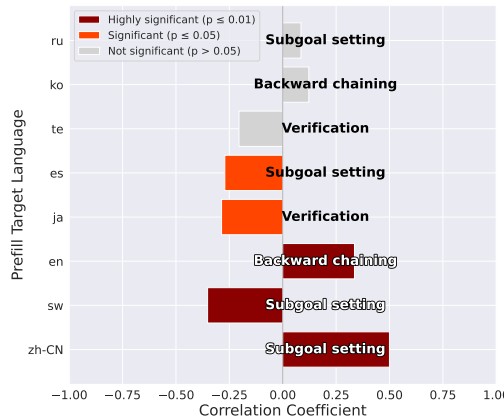

Figure 5: Most correlated habit per language; colors = significance.

**Classification.** Each step is then assigned to one of four theoretically grounded habits (Gandhi et al., 2025):

- **Subgoal setting:** breaking the problem into intermediate goals (e.g., "First, I'll try...").
- **Backtracking:** discarding an incorrect path (e.g., "Let me try again...").
- **Verification:** checking correctness (e.g., "Let's verify this calculation").
- **Backward chaining:** reasoning backward from the target (e.g., "If we want 42, then we need...").

To avoid distortion, we allow an *Others* category, capturing genuinely novel patterns (Appx. A.4.2).

### 6.2 REASONING BEHAVIOURS AND PREFILL LANGUAGE EFFECTS

**Correlational findings.** Aggregating across four models, we computed Pearson correlations between habit counts and accuracy per input × prefill setting. Figure 4 shows that prefill languages reliably shift reasoning styles: Chinese ↑ *Subgoal setting* ($r = 0.50$, $p < .001$) and *Verification* ($r = 0.41$, $p < .001$); Swahili ↓ *Subgoal setting* ($r = -0.35$, $p < .01$); English ↑ *Backward chaining* ($r = 0.30$, $p < .015$).

**Link to performance.** Subgoal-heavy traces (predominantly Chinese-prefilled) yielded $+7.3\%$ higher accuracy on MATH-500. Bootstrapped 95% CIs confirm this is statistically reliable, though modest. Figure 5 highlights the most correlated habit per language.

**Interpretation and caveats.** We hypothesize linguistic anchors prime different decomposition strategies, consistent with bilingual problem-solving studies (Bernardo & Calleja, 2005). However, these results are correlational; causal factors such as tokenization or alignment artifacts cannot be excluded. Controlled interventions with synthetic prefills are a promising next step.

## 7 CONCLUSION

In this work, we reveal that LRMs, despite their strong multilingual ability, predominantly still prefer to reason in hub languages such as English, regardless of the input language. Our introduction of a text pre-filling method provides a practical approach to guide the reasoning language with high success. We demonstrated an asymmetric effect: forcing models to reason in non-hub languages degrades performance in low-resource languages, whereas aligning reasoning with hub languages improves or maintains the performance in reasoning tasks. However, in the cultural reasoning task, native-language reasoning can be beneficial. These findings underscore the critical importance of considering the internal reasoning language to be more inclusive for future models.

## ETHICS STATEMENT

The deployment of Large Reasoning Models across multilingual contexts reveals systematic biases that create substantial inequities in AI access and performance. Our findings demonstrate that LRMs default to reasoning in high-resource languages like English regardless of input language, causing accuracy drops of up to 30 percentage points for speakers of low-resource languages such as Swahili and Telugu on reasoning tasks. This disparity means that billions of speakers of non-dominant languages receive demonstrably inferior service from AI systems, potentially limiting their access to educational tools, professional services, and economic opportunities that increasingly rely on AI capabilities. As reasoning models become more prevalent in critical applications, including education, healthcare, and financial services, these performance gaps risk creating a two-tiered system where advanced AI capabilities remain accessible primarily to speakers of dominant languages, thereby exacerbating existing global inequalities. Furthermore, when models expose their reasoning traces for transparency, users encounter explanations in languages they may not understand, undermining the very trust these features aim to build. To address these concerns, we release our evaluation framework publicly to enable systematic assessment of multilingual disparities and emphasize the urgent need for development practices that ensure equitable performance across all languages rather than optimizing solely for dominant language communities.

## REPRODUCIBILITY STATEMENT

All experiments for evaluating the change of reasoning languages were conducted on workstation of 3 x RTX 3090 GPUs. For smaller models such as DeepSeek-Qwen-7B, DeepSeek-Qwen-1.5B, DeepSeek-Llama-8B we uses sglang inference library (v0.47) for inference and for other models we uses 4 x H100 GPUs from Google Cloud Platform. For fairness, all models are evaluated using the recommended parameters ( temperature, top-p ) from their papers or official releases. Approximately $800 USD was spent on cloud inference. All the code, dataset will be released on publicly accessible platforms (e.g., GitHub, HuggingFace).

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

## A  APPENDIX

### A.1  USE OF LLM IN WRITING

We used ChatGPT (gpt-5), Claude (Sonnet 4, Opus 4.1) and Gemini (Gemini 2.5 Pro) to improve the writing, mainly on grammar fix and use to propose fix on in correct latex table syntax.

Table 7: Decoding parameters used for each model during evaluation.

| Model | Temperature | Top-$p$ | Top-$k$ | Min-$p$ |
|---|---|---|---|---|
| DeepSeek-R1-Distill-Llama-8B | 0.6 | 0.95 | — | — |
| DeepSeek-R1-Distill-Qwen-14B | 0.6 | 0.95 | — | — |
| Qwen3-30B-A3B (reasoning on / off) | 0.6 | 0.95 | 20 | 0 |
| QwQ-32B | 0.6 | 0.95 | — | 0 |

## A.2  MODEL DETAILS

We use the latest sglang inference engine to evaluate all open weights model on A100 GPU with the exception of QwQ-32B which uses Together.ai serverless API endpoint.

As of the decoding parameters we used for all models which was recommended by original model provider Table 7.

For the base model experiments found in Table 17, we simply set temperature = 0.6 only.

## A.3  INFERENCE COST

QwQ-32B cost around 600 USD for all the experiments including ablation studies in scaling efficiency. While other models: Deepseek-Distill-Qwen-14B, Deepseek-Distill-Llama-8B, Qwen3-30B-A3B cost around 1,200 USD in A100 GPUs cost calculated at 1.8 USD per hour per card.

The entire inference process took over 2 weeks to finish under 2 A100 GPUs, using the latest sglang inference service.

Table 8: Data Count Distribution Across Models

| Model | Count |
| --- | --- |
| deepseek-r1-zero | 647 |
| meta math | 539 |
| gemini-flash-thinking | 530 |
| deepseek-r1 | 517 |
| qwq-preview | 506 |
| metamath-qwen | 402 |
| openr1-preview | 116 |
| claude-3-7 | 47 |

Table 9: Training Parameters for ModernBERT-large

| Parameter | Best | Searched |
| --- | --- | --- |
| Learning Rate | $8 \times 10^{-5}$ | $\{5 \times 10^{-5}, 8 \times 10^{-5}, 1 \times 10^{-4}, 3 \times 10^{-4}\}$ |
| Batch Size | 24 | $\{16, 24, 32\}$ |
| Weight Decay | 0.01 | - |
| Number of Epochs | 10 | - |
| Warmup Steps | 50 | - |
| Optimizer | AdamW | - |

## A.4 REASONING PROCESS ANALYSIS

### A.4.1 SEGMENTATION DETAILS

In this section we provided the details we used to curate dataset and the training our segmentation model.

**Dataset**   We collect existing reasoning dataset shared by others from huggingface. We mainly collect reasoning process from Deepseek-R1, Deepseek-R1-Zero, Gemini-2.0-Flash, Claude-3-7-Sonnet, QwQ-preview, MetaMath CoT response (Yu et al., 2023) and Open-R1 : an attempt to generate long CoT from Qwen models. The amount of reasonings from each models can be found in Table 8. For each reasoning, we prompt gpt-4o-2024-07-18 with 1-shot segmentation prompt to segment the reasoning text into steps. Prompts can be found in Figure 7. The raw output is then processed into a sequence chunk which we can used to train a small segmentation model. The annotation cost around 35 USD without any batch discount.

**Hyperparameters**   We split the dataset into 7:3 train and validation set. And we simply use the validation to select the best hyperparameters as found in Table 9 which achieve a high F1 score of 96.08. Training a single hyperparameters took around 4 hours to finished on 4090 GPU.

**Inputs and Target Formats**   Figure 6 illustrates the ModernBERT segmentation process. For each thinking process extracted from model responses, we first split the text by newline symbols, replacing each with a special token (<sep>). The model is trained to predict whether each <sep> token indicates the beginning of a new reasoning step (1) or the continuation of the current step (0). As shown in the figure, ModernBERT takes a reasoning sequence as input (top) and processes mathematical expressions (x + y = 5, y = 5 - x, z + y = 10), classifying each separator position to enable structured parsing of complex reasoning chains. This binary classification approach allows the model to effectively identify logical breakpoints in reasoning processes.

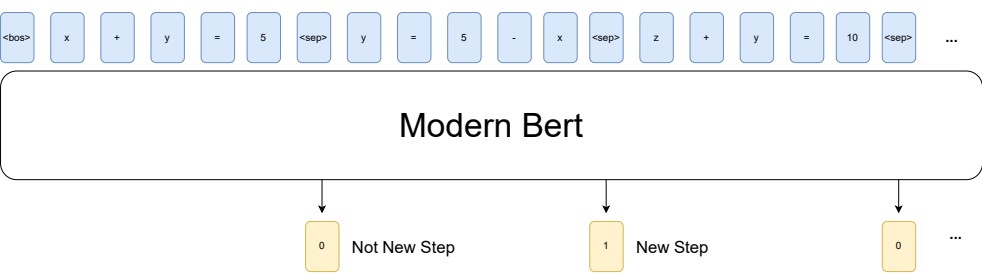

Figure 6: A showcase of how segmentation prediction works

Output your segmentation result by adding a <sep> to the original text to indicate a separation between steps

Do not modify the original reasoning text, only add a separation token

Do not split table into segments, keep a whole table as one step

# Example

```
[INPUT]:
```
Okay, let's see. ...
Alright, let's break this down. First, ...

**Final Answer**
```
\boxed{251.60}
```

```
[OUTPUT]:
```
Okay, let's see. So ...
<sep>
Alright, let's break this down. ...
```
[Skip for brevity]
```
...
<sep>
**Final Answer**
```
\boxed{251.60}
```

Now do the same task by following the same pattern as above:

```
[INPUT]:
```
```
thinking process goes here
```

```
[OUTPUT]:
```

Figure 7: The prompt template for segmenting reasoning steps with <sep> tokens.

### A.4.2 REASONING PROCESS CLASSIFICATION

After segmentation, we concatenate the individual reasoning processes using numbered step tokens (e.g., <step_1>reasoning process 1 <step_1>\n <step_2>reasoning process 2 <step_2>...). This structured sequence, along with the original question, is then passed to a classification prompt as illustrated in Figure 8. We utilize gemini-2.0-flash to perform the classification of each reasoning step according to our taxonomy.

While we initially explored more sophisticated taxonomies that included problem reading and abduction classification, the complexity of these frameworks exceeded the classification capabilities of current LLMs, limiting potential downstream insights. We therefore opted for the simpler four-habits taxonomy. Investigating more complex taxonomies remains an avenue for future research.

---

Here is a problem and the reasoning process that an LLM generated when it tries to solve the problem.

Problem: (enclosed in double backticks)
"
problem
"

Reasoning process: (enclosed in triple backticks, the reasoning process has been split into distinct reasoning steps in the format of <step_idx><reasoning_step_content></step_idx>)
```
reasoning
```

Your task is to classify each reasoning step into one of the following reasoning types: (specified by <type_index>. <type_name>: <definition>)
1. Subgoal setting: Where the model breaks down the problem into smaller, intermediate goals (e.g., 'To solve this, we first need to...' or 'First, I'll try to ..., then ...'
2. Backtracking: Where the model realizes a path won't work and explicitly goes back to try a different approach. An example of backtracking is: 'Let me try again' or 'we need to try a different approach'.
3. Verification: Where the model checks the correctness of the intermediate results or to make sure the final answer is correct.
4. Backward chaining: Where the model works backward from its answer to see whether it can derive the variables in the original problem.
5. Others: This reasoning step is the continuation of the previous reasoning step, or it does not fall into any of the above categories.

Generate the rationale before you make the classification. Provide your output in the following format:

[Reasoning]

<step_1><rationale_1><type_name_1></step_1>
<step_2><rationale_2><type_name_2></step_2>
...
[Final answer]

<step_1><type_name_1></step_1>
<step_2><type_name_2></step_2>
...

---

Figure 8: The prompt template for the classifying each steps into four habits classes.

Table 10: Comparison between Counting (Gandhi et al., 2025) and **seg-class (ours)** methods for `R1-Distill-Llama-8B` on MATH-500 benchmark (English problem statements; generation prefixed with target language)

| Lang | Subgoal setting | | Backtracking | | Verification | | Backward chaining | |
| --- | --- | --- | --- | --- | --- | --- | --- | --- |
| | Count | seg-class | Count | seg-class | Count | seg-class | Count | seg-class |
| **En** | 6.02 | **2.73** | 4.66 | **0.76** | 6.90 | **7.27** | 2.45 | **0.016** |
| **Zh** | 6.83 | **3.26** | 5.89 | **0.65** | 7.76 | **8.45** | 2.49 | **0.018** |
| **Es** | 3.67 | **1.81** | 0.76 | **0.18** | 1.61 | **0.34** | 0.48 | **0.0** |
| **Ru** | 6.46 | **2.84** | 5.27 | **0.84** | 6.67 | **5.34** | 2.87 | **0.006** |
| **Ja** | 5.08 | **2.97** | 1.87 | **0.60** | 8.53 | **3.58** | 0.81 | **0.004** |
| **Ko** | 5.29 | **2.36** | 2.58 | **0.39** | 4.82 | **5.06** | 1.65 | **0.006** |
| **Te** | 2.67 | **0.68** | 1.29 | **0.17** | 2.08 | **1.11** | 1.36 | **0.0** |
| **Sw** | 4.62 | **1.32** | 1.51 | **0.23** | 4.07 | **1.33** | 1.58 | **0.011** |

### A.4.3 COMPARISON OF SEGMENTATION-CLASSIFICATION AND PROMPT-BASE COUNTING METHOD

In this section, we showcase the behavior calculated by the prior work Gandhi et al. (2025) using counting prompt and compared to our segmentation-classification method (seg-class). As seen in the results, our result always resulted in lower behavior numbers than Counting method.

Table 11: Most Frequent Starting Phrases by Model and Language, (-) indicate using the most common prefill target phrase from other models.

| Model | Language | Most Frequent Phrase | Count | Representative Phrase (Count) |
|---|---|---|---|---|
| R1-Distill-Llama-8B | es | Okay | 248 | Primero (224) |
| R1-Distill-Llama-8B | sw | Okay | 253 | Mama (62) |
| R1-Distill-Llama-8B | en | Okay | 451 | Okay (451) |
| R1-Distill-Llama-8B | ja | 好,我现在要 | 196 | まず (112) |
| R1-Distill-Llama-8B | ko | 首先,我需要 | 107 | 먼저 (35) |
| R1-Distill-Llama-8B | ru | Хорошо | 130 | Хорошо (130) |
| R1-Distill-Llama-8B | zh-CN | 嗯 | 305 | 嗯 (305) |
| Qwen-14B | es | Okay, | 209 | Primero (166) |
| Qwen-14B | sw | Okay, so I | 173 | Kwa (43) |
| Qwen-14B | en | Okay, | 345 | Okay (345) |
| Qwen-14B | ja | 好, | 204 | まず (150) |
| Qwen-14B | ko | 嗯, | 204 | 먼저 (78) |
| Qwen-14B | ru | Хорошо | 278 | Хорошо (386) |
| Qwen-14B | zh-CN | 首先, | 181 | 首先 (181) |
| QwQ-32B | es | Okay, | 489 | Primero (3) |
| QwQ-32B | sw | Okay, | 474 | Ili kup (2) |
| QwQ-32B | en | Okay, | 477 | Okay, (477) |
| QwQ-32B | ja | Alright, | 208 | まず (123) |
| QwQ-32B | ko | 좋아 | 220 | 좋아 (220) |
| QwQ-32B | ru | Хорошо | 365 | Хорошо (365) |
| QwQ-32B | zh-CN | 嗯, | 479 | 嗯, (479) |
| QwQ-32B | te | Okay, | 499 | prārambhimcaḍāniki (-) |
| Qwen3-30B-A3B | es | Okay, | 490 | Primero (5) |
| Qwen3-30B-A3B | sw | Okay, | 494 | Ili kup (1) |
| Qwen3-30B-A3B | en | Okay, | 487 | Okay, (487) |
| Qwen3-30B-A3B | ja | Okay, | 491 | まず (5) |
| Qwen3-30B-A3B | te | Okay, | 499 | prārambhimcaḍāniki (3) |
| Qwen3-30B-A3B | ko | Okay, | 491 | 좋아 (2) |
| Qwen3-30B-A3B | ru | Хорошо | 490 | Хорошо (490) |
| Qwen3-30B-A3B | zh-CN | 嗯, | 487 | 嗯, (487) |

## A.5 Prefill Phrases

### A.5.1 Distribution Found From MATH-500 Baseline

To find the distribution of prefill tokens across different languages and models, we analyzed the output generations from multiple language models on a subset of the MATH-500 baseline dataset. For each model and language combination, we recorded the first n tokens generated (where n=4 in our analysis) and tracked their frequencies across all sampled problems.

We implemented a token tracking system that builds up sequences by concatenating successive tokens (e.g., first token, first+second tokens, etc.) and maintains frequency counts for each unique sequence at each position. For models where we had access to the tokenizer, we performed additional analysis by converting between token IDs and human-readable text, allowing us to identify meaningful phrases rather than just token sequences. This double decoding process was particularly valuable for non-Latin script languages where token boundaries might not align with linguistic units. The resulting distributions, shown in Table 11.

## A.6 CULTURALBENCH PREFILL PHRASE

The following Table 12 showcase the phrases used to prefill target language in CultureBench-Hard.

Table 12: Preferred prefill tokens used by language models across different countries, reflecting culturally-specific conversational cues.

| Country | Prefill token |
|---|---|
| Argentina | Vale |
| Australia | Okay |
| Brazil | Tudo bem |
| Canada | Okay |
| Chile | Vale |
| China | 嗯 |
| Czech Republic | Dobře |
| France | D'accord |
| Germany | In Ordnung |
| Hong Kong | 嗯 |
| Indonesia | Baiklah |
| Italy | Va bene |
| Japan | まず |
| Malaysia | Baiklah |
| Mexico | Órale |
| Netherlands | Oké |
| New Zealand | Okay |
| Nigeria | Okay |
| Peru | Ya |
| Philippines | Sige |
| Poland | Dobrze |
| Romania | Bine |
| Russia | Хорошо |
| Singapore | Okay |
| South Africa | Okay |
| South Korea | 먼저 |
| Spain | Vale |
| Taiwan | 嗯 |
| Turkey | Tamam |
| Ukraine | Добре |
| United Kingdom | Alright |
| United States | Okay |
| Zimbabwe | Okay |

Table 13: Comparison of English vs. Native prefill strategies on MATH-500 across languages order by speakers population.

| Strategy | Chinese | Spanish | Russian | Swahili | Japanese | Telugu | Korean |
|---|---|---|---|---|---|---|---|
| **DeepSeek-Qwen-1.5B** | | | | | | | |
| English Prefill | 73.6% | 73.2% | 71.8% | 26.0% | 68.4% | 44.4% | 62.6% |
| Native Prefill | 72.4% | 54.4% | 56.8% | 8.0% | 37.4% | 8.0% | 36.6% |
| Baseline | 69.2 % | 68.0% | 58.8% | 26.2% | 61.0% | 39.6% | 55.2% |
| Difference (EN - Native) | +1.2% | **+18.8%** | **+15.0%** | **+18.0%** | **+31.0%** | **+36.4%** | **+26.0%** |
| **DeepSeek-Qwen-7B** | | | | | | | |
| English Prefill | 82.6% | 85.6% | 85.2% | 37.2% | 82.8% | 67.6% | 81.8% |
| Native Prefill | 85.6% | 68.4% | 79.0% | 21.8% | 56.8% | 34.6% | 53.8% |
| Baseline | 87% | 84.6% | 83.2% | 36.4% | 78.2% | 61.8% | 80.2% |
| Difference (EN - Native) | -3.0% | **+17.2%** | **+6.2%** | **+15.4%** | **+26.0%** | **+33.0%** | **+28.0%** |
| **DeepSeek-Qwen-14B** | | | | | | | |
| English Prefill | 88.4% | 88.6% | 86.6% | 52.4% | 85.2% | 66.2% | 84.4% |
| Native Prefill | 89.8% | 66.4% | 86.4% | 14.6% | 63.6% | 34.4% | 83.8% |
| Baseline | 82.6% | 88.0% | 84.6% | 39.8% | 80.0% | 64.6% | 83.4% |
| Difference (EN - Native) | -1.4% | **+22.2%** | **+0.2%** | **+37.8%** | **+21.6%** | **+31.8%** | **+0.6%** |
| **DeepSeek-Llama-8B** | | | | | | | |
| English Prefill | 78.8% | 80.2% | 78.4% | 37.0% | 74.6% | 42.2% | 69.8% |
| Native Prefill | 73.6% | 45.6% | 59.4% | 3.8% | 32.6% | 16.8% | 41.6% |
| Baseline | 75.8% | 70.6% | 69.8% | 27.3% | 61.2% | 41.6% | 64.6% |
| Difference (EN - Native) | **+5.2%** | **+34.6%** | **+19.0%** | **+33.2%** | **+42.0%** | **+25.4%** | **+28.2%** |
| **DeepSeek-Llama-70B** | | | | | | | |
| English Prefill | 87.0% | 89.2% | 88.8% | 81.8% | 87.4% | 85.4% | 85.6% |
| Native Prefill | 89.0% | 71.4% | 85.8% | 66.8% | 68.8% | 68.4% | 70.8% |
| Baseline | 87.8% | 88.6% | 85.8% | 75.4% | 84.8% | 76.0% | 83.8% |
| Difference (EN - Native) | -2.0% | **+17.8%** | **+3.0%** | **+15.0%** | **+18.6%** | **+17.0%** | **+14.8%** |
| **QwQ-32B** | | | | | | | |
| English Prefill | 92.4% | 92.2% | 91.2% | 67.8% | 90.2% | 84.4% | 90.6% |
| Native Prefill | 90.6% | 93.2% | 90.6% | 55.6% | 87.4% | 65.2% | 88.2% |
| Baseline | 90.8% | 93.2% | 90.8% | 68.2% | 89.4% | 85.0% | 89.0% |
| Difference (EN - Native) | **+1.8%** | -1.0% | **+0.6%** | **+12.2%** | **+2.8%** | **+19.2%** | **+2.4%** |
| **Qwen3-30B-A3B** | | | | | | | |
| English Prefill | 91.4% | 91.0% | 90.6% | 72.4% | 89.4% | 87.0% | 89.8% |
| Native Prefill | 89.4% | 83.8% | 90.0% | 29.6% | 81.8% | 68.4% | 88.0% |
| Baseline | 89.4% | 91.2% | 90.4% | 72.8% | 90.0% | 87.7% | 89.2% |
| Difference (EN - Native) | **+2.0%** | **+7.2%** | **+0.6%** | **+42.8%** | **+7.6%** | **+18.6%** | **+1.8%** |
| **Average across all models** | | | | | | | |
| English Prefill | 84.9% | 85.7% | 84.7% | 53.5% | 82.6% | 68.2% | 80.6% |
| Native Prefill | 84.4% | 69.0% | 78.3% | 28.6% | 61.2% | 42.3% | 66.1% |
| Baseline | 83.3% | 83.5% | 80.5% | 49.4% | 77.8% | 65.2% | 77.9% |
| Difference (EN - Native) | **+0.6%** | **+16.7%** | **+6.3%** | **+24.9%** | **+21.3%** | **+25.9%** | **+14.5%** |

A.7 ADDITIONAL MATH-500 DETAILS

Table 13 shows the scores of all models with the inclusion of baseline score.

Table 14: Comparison of MMLU performance when reasoning in native language vs. English

| Strategy | English | Chinese | Spanish | Swahili | Japanese | Korean |
|---|---|---|---|---|---|---|
| **DeepSeek-Llama-8B** | | | | | | |
| Prefill English | – | 69.8% | 71.4% | 29.8% | 65.3% | 61.5% |
| Prefill target Language | 67.7% | 63.4% | 53.8% | 18.6% | 46.2% | 46.8% |
| Difference (EN - Native) | – | **+6.4%** | **+17.6%** | **+11.2%** | **+19.1%** | **+14.6%** |
| **DeepSeek-Qwen-14B** | | | | | | |
| Prefill English | – | 84.7% | 85.7% | 44.5% | 83.3% | 81.1% |
| Prefill target Language | 87.3% | 83.3% | 85.8% | 36.4% | 77.3% | 73.4% |
| Difference (EN - Native) | – | **+1.4%** | -0.1% | **+8.1%** | **+5.9%** | **+7.7%** |
| **QwQ-32B** | | | | | | |
| Prefill English | – | 88.7% | 89.1% | 59.8% | 87.8% | 85.8% |
| Prefill target Language | 91.4% | 88.5% | 89.2% | 23.8% | 88.3% | 83.6% |
| Difference (EN - Native) | – | **+0.3%** | -0.1% | **+36.0%** | -0.5% | **+2.2%** |
| **Qwen3-30B-A3B** | | | | | | |
| Prefill English | – | 88.9% | 89.0% | 61.1% | 86.8% | 82.1% |
| Prefill target Language | 85.4% | 85.6% | 86.0% | 62.2% | 84.2% | 81.0% |
| Difference (EN - Native) | – | **+3.3%** | **+3.0%** | -1.0% | **+2.7%** | **+1.1%** |
| **Average across all models** | | | | | | |
| Prefill English | – | 83.1% | 83.8% | 48.8% | 80.8% | 77.6% |
| Prefill target Language | 83.0% | 80.2% | 78.7% | 35.3% | 74.0% | 71.2% |
| Difference (EN - Native) | – | **+2.9%** | **+5.1%** | **+13.6%** | **+6.8%** | **+6.4%** |

Table 15: Comparison of MMMLU partial (subset) and full accuracy scores across different models and language configurations.

| Model | Input | Reasoning | Partial Acc. | Full Acc. | Diff. |
|---|---|---|---|---|---|
| QwQ-32B | en | es | 91.04% | 88.52% | +2.52% |
| DeepSeek-Qwen-14B | en | en | 88.02% | 85.61% | +2.41% |
| DeepSeek-Qwen-14B | en | zh-CN | 86.63% | 84.09% | +2.54% |
| DeepSeek-Qwen-14B | en | ko | 85.40% | 82.52% | +2.88% |
| DeepSeek-Qwen-14B | es | en | 85.69% | 82.68% | +3.01% |
| DeepSeek-Qwen-14B | en | es | 84.63% | 82.12% | +2.51% |
| DeepSeek-Qwen-14B | en | ja | 83.74% | 81.29% | +2.45% |
| DeepSeek-Qwen-14B | zh-CN | zh-CN | 83.64% | 80.33% | +3.31% |
| DeepSeek-Qwen-14B | ja | en | 83.26% | 79.97% | +3.29% |
| DeepSeek-Qwen-14B | ko | en | 81.08% | 78.02% | +3.06% |

## A.8 MMMLU RESULTS

### A.8.1 MMMLU FULL MODELS BREAKDOWN

Table 14 shows the full results for four models. We observe a significant jump in QwQ-32B where switching Swahili MMLU from English reasoning to Swahili reasoning drops by over 36%.

### A.8.2 SCORES IN SUBSET VERSUS FULL SET

Table 15 showcases the accuracy between the 32 subjects and the full 56 subjects score. All settings consistently score higher than the full set; however, the correlation score between different settings is 0.9953 with a p-value lower than 0.0001. This means the subsets we have chosen are representative enough of the full MMMLU test set.

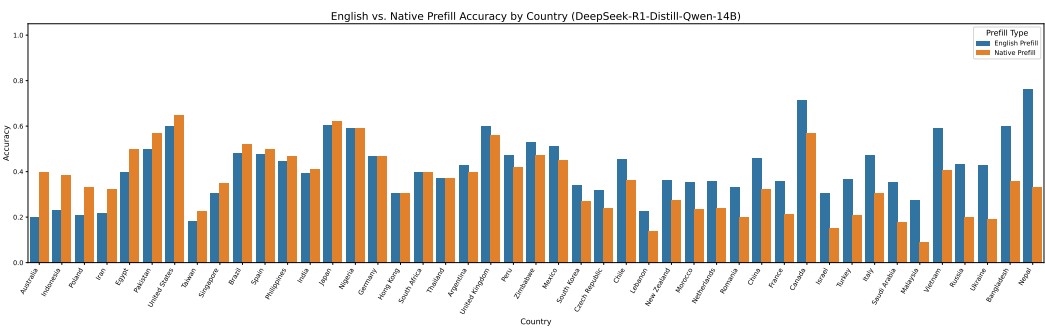

Figure 9: Sorted by positive improvements from using native language reasoning compare to english reasoning in Deepseek-Qwen-14B

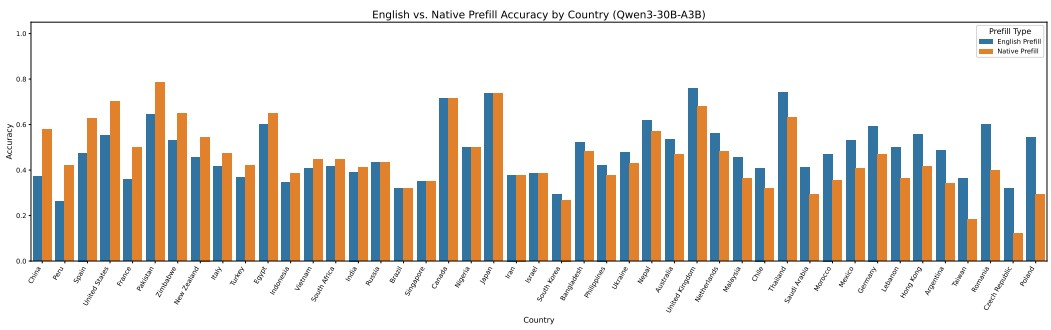

Figure 10: Sorted by positive improvements from using native language reasoning compare to english reasoning in Qwen3-30B-A3B

## A.9 CULTURALBENCH RESULTS

In CulturalBench, we maintained the original English questions while only varying the reasoning language. This approach preserves the precise wording of questions, as translation could potentially compromise the cultural nuances embedded in specific English terminology unique to each culture.

Figures 9, 10, and 11 illustrate the performance difference between using English prefills versus prefills in the predominant language of each respective country.

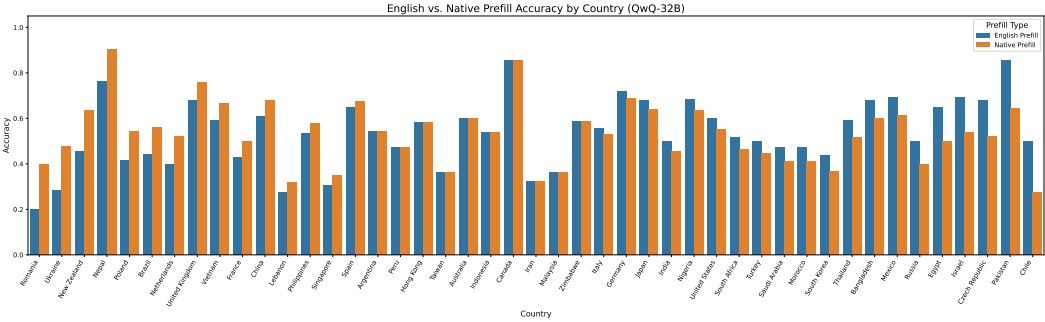

Figure 11: Sorted by positive improvements from using native language reasoning compare to english reasoning in QwQ-32B

Table 16: Comparison of LMSYS-Toxic ASR score when reasoning in English vs. the target language, across languages ordered by speakers' population.

| Strategy | Chinese | Spanish | Russian | Swahili | Japanese | Telugu | Korean |
|---|---|---|---|---|---|---|---|
| **DeepSeek-Llama-8B** | | | | | | | |
| Prefill English (EN) | 15.0% | 27.3% | 21.3% | 3.4% | 18.4% | 0.8% | 7.1% |
| Prefill Target Language | 18.9% | 37.6% | 32.9% | 3.9% | 20.4% | 2.3% | 5.9% |
| Difference (EN - Target) | -3.8% | -10.3% | -11.6% | -0.5% | -2.0% | -1.6% | +1.2% |
| **DeepSeek-Qwen-1.5B** | | | | | | | |
| Prefill English (EN) | 17.31% | 20.51% | 13.38% | 3.26% | 7.72% | 0.00% | 2.18% |
| Prefill Target Language | 19.29% | 19.89% | 17.59% | 2.95% | 5.90% | 0.00% | 1.45% |
| Difference (EN - Target) | -1.98% | 0.62% | -4.21% | 0.31% | 1.82% | 0.00% | 0.73% |
| **DeepSeek-Qwen-7B** | | | | | | | |
| Prefill English (EN) | 13.58% | 16.48% | 15.71% | 4.13% | 11.21% | 0.00% | 6.12% |
| Prefill Target Language | 13.87% | 14.02% | 14.33% | 3.32% | 9.51% | 0.00% | 4.55% |
| Difference (EN - Target) | -0.29% | 2.46% | 1.38% | 0.81% | 1.70% | 0.00% | 1.58% |
| **DeepSeek-Qwen-14B** | | | | | | | |
| Prefill English (EN) | 7.0% | 8.0% | 9.0% | 4.5% | 7.8% | 0.6% | 4.1% |
| Prefill Target Language | 5.0% | 7.6% | 12.6% | 4.3% | 8.2% | 2.1% | 3.5% |
| Difference (EN - Target) | +2.0% | +0.3% | -3.7% | +0.1% | -0.3% | -1.5% | +0.6% |
| **QwQ-32B** | | | | | | | |
| Prefill English (EN) | 4.1% | 6.6% | 6.4% | 3.1% | 6.6% | 1.1% | 2.9% |
| Prefill Target Language | 3.2% | 3.1% | 8.1% | 1.1% | 4.9% | 0.6% | 2.4% |
| Difference (EN - Target) | +0.9% | +3.5% | -1.7% | +2.0% | +1.7% | +0.4% | +0.5% |
| **Qwen3-30B-A3B** | | | | | | | |
| Prefill English (EN) | 4.6% | 7.1% | 9.1% | 2.9% | 6.9% | 0.6% | 4.1% |
| Prefill Target Language | 2.6% | 4.8% | 10.5% | 5.1% | 4.4% | 1.2% | 3.3% |
| Difference (EN - Target) | +2.0% | +2.3% | -1.4% | -2.1% | +2.5% | -0.6% | +0.9% |
| **Average across all models** | | | | | | | |
| Prefill English (EN) | 10.3% | 14.3% | 12.5% | 3.6% | 9.8% | 0.5% | 4.4% |
| Prefill Target Language | 10.5% | 14.5% | 16.0% | 3.4% | 8.9% | 1.0% | 3.5% |
| Difference (EN - Target) | -0.2% | -0.2% | -3.5% | +0.1% | +0.9% | -0.5% | +0.9% |

## A.10 ADDITIONAL LMSYS-TOXIC BENCHMARKS DETAILS

In LMSYS-Toxic, we observed that RL-finetuned model QwQ-32B resulted in lower attack success rate (ASR) when reasoning in their native language for most non-English languages (Japanese, Korean, Chinese, Spanish), with the notable exception of Russian. As shown in Table 16, QwQ-32B and Qwen3 models demonstrate a consistent pattern where forcing English reasoning (via "Okay" prefilling) increases toxicity rates by 1-3.5 percentage points for Japanese, Korean, Chinese, and Spanish inputs. Interestingly, the Russian language exhibits the opposite pattern, with lower toxicity when reasoning is guided toward English rather than maintaining native Russian reasoning.

Table 17: AIME-24 pass@8 from Qwen3-30B-A3B **base** model with different initial phrase for text completion.

| | Language | | | | | |
|---|---|---|---|---|---|---|
| | en | zh-CN | ja | ru | ko | sw |
| Phrase | Okay | 嗯 | まず | Хорошо | 먼저 | Kwa kuzingatia |
| pass@8 | **0.267** | 0.190 | 0.172 | 0.133 | 0.133 | 0.200 |

## A.11 STUDY OF IMPACT OF PREFILL TOKENS IN PRETRAINED MODEL

To investigate why models might gravitate towards English and Chinese for reasoning, we conducted an experiment using a small mathematics problem set, AIME-2024. Using the prompt template from Deepseek-R1-zero (Guo et al., 2025), we prompted the Qwen3-30B-A3B base model (without post-training) in a zero-shot pass@8 setting. To encourage reasoning in languages other than English, we prepended an initial phrase in the target language to the prompt, guiding the model to complete its reasoning in that language. The results, presented in Table 17, show that English-led reasoning significantly outperforms other languages for this base model.

Based on these findings, we hypothesize that during the RL training phase, models tend to exploit the language that allows the most effective CoT generation to maximize the final task score. Since the choice of reasoning language is typically not an explicit part of the reward function, leveraging the language in which the underlying base model performs best (as suggested by Table 17 for English) becomes an optimal strategy for achieving higher rewards.

Table 18: Comparison prefilling reasoning chain with native language or english in reasoning on, while prefilling the response in reasoning off, meaning the model does not undergo long CoT process before output response.

| MATH-500 | Language | | | | | |
|---|---|---|---|---|---|---|
| Model Configuration | Chinese | Japanese | Korean | Spanish | Russian | Telugu |
| *Qwen3 30 A3B (reasoning off)* | | | | | | |
| Prefill English (To evaluate) | 84.8% | 81.6% | 80.0% | 80.2% | 82.2% | 81.2% |
| Prefill Input Language | 88.4% | 80.8% | 82.2% | 81.2% | 79.4% | 68.3% |
| Difference (English - Input) | -3.6% | 0.8% | -2.2% | 1.0% | 2.8% | 12.9% |
| *Qwen3 30 A3B (reasoning on)* | | | | | | |
| Prefill English (Okay) | 91.4% | 89.4% | 89.8% | 91.0% | 90.6% | 87.0% |
| Prefill Input Language | 89.4% | 81.8% | 88.0% | 83.8% | 90.0% | 68.4% |
| Difference (English - Input) | 2% | 7.6% | 1.8% | 7.2% | 0.6% | 18.6% |

## A.12 BRITTLENESS OF LANGUAGE GUIDANCE IN LRM COMPARED TO TYPICAL COT FOUND IN LLMS

Since Qwen3-30B-A3B allows us to trigger reasoning mode on and off, we first compare the sensitivity between reasoning and normal CoT prompts. Specially we compare the results between prefilling the phrase in reasoning versus prefilling in the response in CoT response with reasoning mode off. Table 18 shows that the penalty of changing reasoning language is far more worse than changing in typical chain of thought from LLMs.

| Dataset | Test Split Size | License |
|---|---|---|
| MMMLU[1] | N = 14,042 (per language) | MIT License |
| CulturalBench-Hard[2] | N = 4,709 | CC-BY-4.0 |
| LMSYS-toxic[3] | N = 2,000 (per language) | LMSYS-Chat-1M Dataset License Agreement |
| MATH-500[4] | N = 500 (per language) | MIT License |

Table 19: AI Dataset Information with Test Split Sizes

### A.13 DATASET DETAILS

Table 19 contains each of the benchmarks and their licenses.

**Languages:**

- MMMLU: English, Spanish, Japanese, Korean, Swahili, Chinese

- CulturalBench-Hard: 30 countries

- LMSYS-toxic: English, Japanese, Spanish, Korean, Swahili, Telugu, Russian, Chinese

- MATH-500: English, Japanese, Korean, Spanish, Swahili, Telugu, Russian, Chinese

**HuggingFace Link:** [1] MMMLU: `https://huggingface.co/datasets/openai/MMMLU`
[2] CulturalBench: `https://huggingface.co/datasets/kellycyy/CulturalBench`
[3] LMsys-Chat-1M: `https://huggingface.co/datasets/lmsys/lmsys-chat-1m`
[4] MATH-500: `https://huggingface.co/datasets/HuggingFaceH4/MATH-500`

Table 20: Correlation between **prefill target languages** and reasoning behaviors

| Language | Backtrack | Backward | Subgoal Setting | Verification |
|----------|-----------|----------|-----------------|--------------|
| English | -0.07 | 0.34** | 0.22 | 0.07 |
| Spanish | 0.08 | -0.16 | $-0.27^*$ | $-0.26^*$ |
| Japanese | 0.16 | -0.19 | -0.19 | $-0.29^*$ |
| Korean | -0.03 | 0.12 | 0.02 | 0.05 |
| Russian | -0.06 | 0.02 | 0.09 | 0.05 |
| Swahili | -0.22 | $-0.25^*$ | $-0.35^{**}$ | 0.16 |
| Telugu | -0.01 | -0.18 | -0.12 | -0.20 |
| zh-CN | 0.23* | 0.02 | 0.50*** | 0.41*** |

$^*p < 0.05$, $^{**}p < 0.01$, $^{***}p < 0.001$

Table 21: Correlation between **input target languages** and reasoning behaviors

| Language | Backtrack | Backward | Subgoal Setting | Verification |
|----------|-----------|----------|-----------------|--------------|
| English | -0.07 | 0.08 | 0.00 | 0.08 |
| Spanish | -0.07 | -0.08 | -0.19 | -0.22 |
| Japanese | 0.14 | 0.01 | -0.14 | -0.19 |
| Korean | -0.06 | 0.18 | 0.04 | 0.03 |
| Russian | -0.10 | 0.00 | 0.09 | 0.07 |
| Swahili | -0.13 | -0.10 | -0.19 | 0.09 |
| Telugu | 0.08 | -0.09 | 0.02 | -0.14 |
| zh-CN | 0.19 | 0.04 | 0.42*** | 0.33** |

$^*p < 0.05$, $^{**}p < 0.01$, $^{***}p < 0.001$

## A.14 BEHAVIOR RESULTS DETAIL FOR MATH-500

This section details the behavioral results observed for the MATH-500 dataset, specifically examining the correlation between language and various reasoning behaviors. The analysis, as presented in Tables 20 and 21, investigates how different languages, when used either as prefill tokens to guide the model's internal "thought" process or as the input language of the problems themselves, influence reasoning strategies such as backtracking, backward chaining, subgoal setting, and verification. Notably, Chinese (zh-CN) prefill tokens show a strong positive correlation with subgoal setting ($r = 0.50, p < 0.001$) and verification ($r = 0.41, p < 0.001$). Conversely, English prefill is significantly positively correlated with backward chaining ($r = 0.34, p < 0.01$), while Swahili shows a significant negative correlation with subgoal setting ($r = -0.35, p < 0.01$) when used as a prefill language. When considering input languages, Chinese again demonstrates a significant positive correlation with subgoal setting ($r = 0.42, p < 0.001$) and verification ($r = 0.33, p < 0.01$). These findings suggest that linguistic context, whether from prefill or input, can systematically influence the reasoning patterns employed by the models when tackling mathematical problems.

