# OpenReview forum: "Language Matters: How Do Multilingual Input and Reasoning Paths Affect Large Reasoning Models?"
_ICLR.cc/2026/Conference — ICLR 2026 Conference Withdrawn Submission_

### Official Review · Reviewer_KLza · 2025-10-22

**Soundness:** 2
**Presentation:** 1
**Contribution:** 2
**Rating:** 2
**Confidence:** 4

**Summary:**

This paper investigates a critical question of which language the models use when solving problems presented in different languages. The experimental results show that reasoning models predominantly default to reasoning in high-resource 'hub' languages such as English, regardless of the input language.

**Strengths:**

- The paper reveals that despite language models' strong multilingual capabilities, they predominantly prefer to reason in hub languages like English. This provides a valuable empirical understanding of the models’ reasoning traces and language biases during reasoning.
- The introduction of a text pre-filling method to steer the reasoning language demonstrates a concrete, effective strategy for improving or maintaining performance in multilingual reasoning tasks, while also highlighting the benefit of native-language reasoning in cultural tasks.

**Weaknesses:**

- The argument and logic in lines 78–85 are difficult to follow. It is unclear how using non-preferred versus preferred languages affects reasoning and non-reasoning tasks, and how the reported asymmetric effect arises.
- The classification of MMMLU as a reasoning benchmark seems questionable. While MMMLU includes some reasoning tasks, it also contains many knowledge-intensive questions (e.g., “In what year did the Continental Reformation begin?”). It would be better to clarify the rationale for treating MMMLU primarily as reasoning tasks or adjust the categorisation accordingly.
- This paper presents unclear methodological descriptions and poorly defined concepts, making it difficult to understand the methods and experimental results. For example, the paper lacks a clear definition and explanation of the internal reasoning process. It is unclear what the “internal language” refers to and how it is obtained.
- The experimental setting is somewhat limited, as most models considered in this paper are predominantly trained on English data, particularly during the post-training phase. Consequently, it is unsurprising that their outputs are biased toward English. Additionally, the analysis remains at a superficial level, focusing mainly on surface outputs, and does not investigate deeper aspects such as cross-lingual knowledge transfer within the internal models.

Others:
- The paper contains inconsistent citation formatting. For example, phrases like “cultural knowledge Chiu et al. (2024)” in line 91 or “(Wu et al., 2024) proposed” in line 117 do not follow standard citation conventions. Please revise all in-text citations to be consistent (e.g., “Wu et al., 2024 proposed …” or “… as shown in previous work (Wu et al., 2024)”).
- The writing could be improved for grammatical consistency and clarity. For example, in lines 143-144 “We adopt … . We selected …”, the tense usage is inconsistent. Please revise for consistent tense and clearer phrasing.
- The organisation of this article is somewhat unreasonable. For example, section 3 appears unbalanced. It begins with extensive descriptions of tasks and datasets but ends with a brief, isolated subsection. The authors may consider reorganising this paper to achieve better structural coherence and balance.
- The use of abbreviations is inconsistent. For example, “LLMs” appears in line 104 before its full form is introduced (line 204), and “CoT” is redefined in lines 202–203 after being introduced earlier (line 99). Please ensure that each abbreviation is defined only once when it first appears and used consistently throughout the paper.
- Overall, the paper suffers from substantial writing and presentation problems that impede readability and evaluation; it requires a careful, comprehensive revision.

**Questions:**

See above Weaknesses.

---

### Official Review · Reviewer_4vri · 2025-10-26

**Soundness:** 2
**Presentation:** 1
**Contribution:** 1
**Rating:** 2
**Confidence:** 5

**Summary:**

This paper performs an evaluation of the reasoning traces in LRMs in eight languages. It finds that LRMs typically reason in English, and that by manipulating the reasoning trace by adding language-specific tokens, certain benchmarks improve while others degrade.

**Strengths:**

1. Multiple languages with a good distribution between low-resource and high-resource
2. Interesting findings on benchmark performance disparity
3. Use of open models
4. Correlation analysis of the results--while not strictly needed as part of the main body, still an excellent thing to see

**Weaknesses:**

This paper has interesting findings and methodology. In my opinion, however, it is not ready to be accepted at ICLR. I believe this to be true due to three major concerns:
1. The writing is not up to par. There are many typos, incomplete sentences, and misuse of \citet versus \citep. Anthropomorphism ('LRMs predominantly think') is rampant.
2. The rigour (both in terms of argumentation and experimentation) is not there. In terms of argumentation, there are several claims (e.g., L195-196; L258) where things that are abstractly quantified ('low resource', 'many previous works', require a source--the latter, in particular, is only using one source). There are well-known works for both LLMs and LRMs showing analogous results that a cursory Google search can find, and are missing in this work. Moreover, dialects are not specified.
3. The contributions mostly point at a problem and an interesting pattern, but do not offer a solution. It is easy to pick an LLM and point at a problem--what else is there?

Minor: the ethics disclosure appears to be the motivation of the work. The ethics section is meant to be a self-evaluation of the implications of your work: could this lead to harm?

Here's how I'd improve this paper:
1. Correct the writing. While some proofreading is certainly required, the main thing missing is the ability to convey and support the work's arguments convincingly and rigorously. This means adding citations, ensuring a good coverage of the existing literature, ensuring abbreviations aren't repeated more than once (abstract is okay), and using more rigour in the description of experiments (see below). Namely, dialects should be specified, especially if grapheme-based (e.g., zh-han-tw versus zh-han-cn).
2. Add measures of statistical significance/error analyses: it is not sufficient to say that language L had an $\pm$X% change in accuracy. What about the labels? Is this a significant increase? What about memorisation--do you believe this played a role?
3. Answer the 'then what' question: you found disparities when inducing reasoning traces in a specific language per benchmark. So what? Why? Then what? This will make your contribution very robust.
4. Last, but not least: avoid anthropomorphism. It is very unscientific and leads to misinterpretation of results. Instead of writing 'in which language do these models _reason_?', you could write something like 'what is the language predominant in the thinking traces?'

I do not think it is ready to be accepted at a venue like ICLR, but if the authors improve upon the existing work I think their contributions could be accepted at a similar venue.

**Questions:**

From above:
1. Do you think memorisation played a role in your results?
2. Which dialects did you use?
3. How can this research be taken further?

---

### Official Review · Reviewer_v6pK · 2025-10-30

**Soundness:** 3
**Presentation:** 3
**Contribution:** 2
**Rating:** 4
**Confidence:** 4

**Summary:**

The paper studies which language large reasoning models (LRMs) "think in" when prompted in different languages, and how forcing the model’s reasoning language affects performance and safety. Across several LRMs, the authors find a strong reasoning hub effect: internal chains-of-thought default to English (and sometimes Chinese) regardless of input language. The paper introduces a simple language-prefill method (e.g., inserting “Okay,” “まず,” “嗯” into the prompt) to steer the reasoning trace language. For reasoning-heavy tasks (MATH-500, MMMLU), forcing English reasoning tends to improve accuracy over native-language reasoning. The gap is largest for lower-resource languages (e.g., Swahili, Telugu). Conversely, on safety and cultural benchmarks (LMSYS-Toxic, CulturalBench-Hard), native-language reasoning can help. The paper also analyzes reasoning habits (subgoal setting, verification, etc.) via a segmentation-classification pipeline, showing that prefills shift habits and modestly correlate with accuracy.

**Strengths:**

The paper focuses on an important question about multilingual reasoning traces. It also proposed a simple prefill method that reliably shifts reasoning language and enables controlled comparisons. The evaluation clearly shows that English-prefilled reasoning often boosts MATH-500/MMMLU, especially for lower-resource languages. In-depth analysis is conducted on scaling and difficulty, demonstrating that harder tasks enlarge the gap and favor English reasoning. However, in the cultural reasoning task, native-language reasoning can be more helpful. Taken together, these findings highlight the importance of the reasoning traces in multilingual LRMs.

**Weaknesses:**

A published paper at EMNLP 2025 [1] also indicates the same findings: LRMs are weak in accuracy when reasoning in non-English languages, compared with English reasoning for multilingual questions. Similarly, "prefix-hacking" (i.e., prefill) prompts are introduced by their paper to force the model to think in a specific language to benefit users speaking different languages. Plus, it also includes a post-training exploration to mitigate the mismatch of reasoning languages when the prompt is in non-English languages. This will diminish the contribution of this paper. Also, it is unclear what metric is used for detecting the language of the reasoning traces.

---

***Reference***:

*[1] When Models Reason in Your Language: Controlling Thinking Language Comes at the Cost of Accuracy. Qi et al., 2025*

**Questions:**

**Novelty over the existing work [1].** Can you clarify which insights are new beyond the trade-off itself and the prefix-hacked prompts (i.e., prefill)? A concise list would help.

**Language detector is unclear.** Sorry if I missed anything, but the paper seems to lack an explicit description of the language detector used for detecting the language of thinking traces, as in [1], they noted that LangDetect was used for the detection.

**Tokenizer controls.** It would be promising to control for tokenizer coverage (e.g., synthetic vocab balancing or byte-level decoding) to test whether hub-ness persists when tokenization disadvantages are removed.

---

***Reference***:

*[1] When Models Reason in Your Language: Controlling Thinking Language Comes at the Cost of Accuracy. Qi et al., 2025*

---

### Official Review · Reviewer_dvyC · 2025-10-31

**Soundness:** 3
**Presentation:** 3
**Contribution:** 2
**Rating:** 4
**Confidence:** 4

**Summary:**

Prior work has shown that LLMs tend to reason internally in a dominant “hub” language, such as English or Chinese, even when given inputs in other languages. This paper extends that finding to Reasoning Models and examines how the choice of reasoning language affects both accuracy and safety. Using a simple language prefill technique that anchors the reasoning language after the "think" token, the authors test several open-weight LRMs across eight languages and multiple benchmarks, including MMMLU, MATH-500, CulturalBench-Hard, and LMSYS-Toxic. They find that models reason most effectively in hub languages, while reasoning in low-resource or native languages reduces accuracy but improves cultural awareness and safety.

**Strengths:**

1. this paper is well written
1. The topic is timely and important, with potential real-world impact—especially for users of low-resource languages who are often overlooked in multilingual LLM research.
1. the authors test across eight diverse languages and several reasoning and behavioral benchmarks, which strengthens the generalizability of the findings.
1. The work provides valuable insights into the trade-off between reasoning performance and cultural or safety alignment

**Weaknesses:**

1. While the study is well executed, prior research has already identified that multilingual models tend to reason in a dominant hub language (e.g., English) and display inconsistencies across languages. Earlier work also attributes this effect partly to decoding and translation issues (models may reason correctly in the hub language internally but fail to translate their reasoning back to the target language). This paper extends that paradigm to reasoning models, but the conceptual advance beyond prior multilingual research in this area feels incremental.


1. The paper is primarily empirical and diagnostic. It documents correlations between reasoning language and performance but does not provide mechanistic insight into why these multilingual inconsistencies arise. Given that the phenomenon itself is not new and has been well studied in non-reasoning models, the work would benefit from a deeper analysis to uncover underlying causes, potential mitigation strategies, or insights specific to reasoning models.

1. Since evaluation is based on the output language, performance differences may still reflect decoding or translation effects rather than true reasoning competence. Ablation studies or multilingual reasoning-path analyses would help disentangle reasoning-language factors.

1. Even when the model is forced to "think" in a target language using a prefill token, its latent reasoning process may still occur in English or another hub language. Prefill alone might not be sufficient to shift the model’s internal reasoning. A deeper representational or attention-trace analysis would help determine whether the observed differences in reasoning language are merely superficial or reflect genuinely distinct reasoning pathways.

1. Most experiments rely on Chinese-origin open-weight models (e.g., DeepSeek, Qwen), which may share similar multilingual training distributions and biases. Including models from different development ecosystems (e.g., LLaMA, Mistral, Gemma) would make the findings more robust and generalizable.

**Questions:**

Pease refer to the weaknesses. I am happy to discuss and would be positive about raising my score during the rebuttal if the authors can provide further explanations and supporting evidence for the main points :)

---

### Note · Authors · 2025-11-19

**Comment:**

We thank all reviewers for the constructive reviews and we've decide to withdraw and refine the paper

**Withdrawal Confirmation:**

I have read and agree with the venue's withdrawal policy on behalf of myself and my co-authors.